# The impact of maternal versus paternal imprisonment on their children's health: A scoping review

**Naomi Gadian**[1]*, **Abigail Dunn**[1], **Donna Arundel**[1], **Sara Morgan**[1], **Paula Harriott**[2], **Lucy Wainwright**[2], **Emma Plugge**[1]

**1** Population Health Sciences Centre, Faculty of Medicine, University of Southampton, Southampton, United Kingdom, **2** EP:IC Consultants, Kent, United Kingdom

* Naomigadian@hotmail.com

## Abstract

### Background

Rates of imprisonment for both women and men are high in England and Wales yet no official records report the number of people in prisons who are parents. Reports suggested 54% of people in prison have children under the age of 18 years which is estimated to affect 312,000 children annually. Research has examined the impact of parental imprisonment on their children, but little is known about the health and well-being outcomes for children who experience maternal versus paternal imprisonment. The Prison Reform Trust reported only 9% of children live with their father at the time of their mother's incarceration, whilst 75% of children live with their mother at their father's incarceration. The aim of this scoping review was to review the published evidence about the health impacts of maternal versus paternal imprisonment to enable a better understanding of the differential impacts on affected children and to identify where gaps in the evidence remain.

### Methods

The Arksey and O'Malley methodology for scoping review was used to address how do the physical, mental and behavioural health, along with healthcare service use differ between children who experience their mother being imprisoned, compared to those who experience their father being imprisoned. Databases searched included Medline, Embase, CINAHL, Cochrane, PyschINFO, Web of Science, Delphis and IBBS. The search yielded 9,773 results, which after screening and removal of duplicates, resulted in 20 papers being included.

which permits unrestricted use, distribution, and reproduction in any medium, provided the original author and source are credited.

**Data availability statement:** The search strategy, exclusion and inclusion criteria are stated in the report. I have included search terms and inclusion /exclusion below. Please do let me know if further detail is required. Search terms Key search terms, including all languages with no timeframe limits, used were imprison* or incarcerat* or jail* or prison* or gaol* Child* or "young person" or adolescen* or teen* or youth* Maternal* or mother* or mum* or mom* or mam* paternal* or father* or dad* or papa* Table 1 in S2 File in the report- Inclusion and exclusion criteria Inclusion Participants: - Any offspring of imprisoned mothers and fathers - Male of female Intervention - Experience of maternal imprisonment or father imprisonment, or both, within the same study population. The comparison group may be experiencing no parental imprisonment Outcome: Any health condition including: Physical health Mental health Behavioural health, for example behavioural challenges and sleeping. The paper must report health outcomes for children who have experienced maternal imprisonment and children who have experienced paternal imprisonment Study: Prevalence studies, cross-sectional studies, cohort studies, case control studies, surveys, Any country, any date of publication, any language Published and un-published data Exclusion: Participants If parent was detained in immigration removal centres or secure psychiatric units or was a prisoner of war Excluded if intervention: Only focused on maternal or only paternal imprisonment Excluded if outcome was Not a health condition, such as income, homeless, school achievement educational level sporting achievement involvement in the criminal justice system family relations Excluded if the study was: Policy, opinion, review articles. (Whilst review articles were excluded, the reference lists were reviewed to check for any missed papers.)

**Funding:** The author(s) received no specific funding for this work.

**Competing interests:** The authors have declared that no competing interests exist.

## Results

All included papers compared data relating to outcomes for children who had experienced maternal or paternal imprisonment to children with no parental imprisonment, and three compared maternal to paternal imprisonment. Eighteen used populations in the United States of America and of these, thirteen used data from two studies. Having experienced either parent being in prison results in considerable impacts on the health of children, as well as their support networks and the stigma they encounter. The findings comprised of four main categories of health: physical health, mental health, behavioural health and healthcare service use.

## Discussion

This review highlighted how atomised the study designs and study populations were in addition to the varied findings about the impact of maternal and paternal imprisonment on children. The sparsity of literature resulted in challenges addressing the original study question about how health and wellbeing outcomes differ for children experiencing maternal versus paternal imprisonment and no clear conclusions can be drawn.

## Conclusion

There is limited understanding about the impact of maternal or paternal imprisonment on their children's health and behaviour, despite the substantial implications their imprisonment has and the stigma. It is important to consider that the absence of clear significant findings, does not negate the great health needs for this cohort. Further research is vital to ensure this population is identified, recognised and supported appropriately.

## Introduction

Rates of imprisonment in England and Wales are high in comparison to many European countries, as shown by one hundred and fifty-four people per one hundred thousand population being imprisoned in England and Wales. Despite the number of people imprisoned in England and Wales marginally falling during the Covid-19 pandemic largely consequential to court pandemic restrictions, in 2020 the annual prison population was estimated to be 80,366 people [1].

It is known that imprisonment can significantly affect the health and wellbeing of those imprisoned, but it can also affect the health and wellbeing of their children. This is especially so where the imprisoned parent has sole caregiving responsibility, or if the parent and child have a particularly strong bond [1–3]. This is important since the number of parents in prison is high. A report published by Crest Advisory, in combination with The Centre for Health and Justice at the Institute of Mental Health University

of Nottingham, estimated that the majority of people who are imprisoned are parents, and 54% have children under the age of 18.

Children who experience parental imprisonment have been referred to as being an invisible group or forgotten victims who are being punished and there is no formal data collection in England [3,4]. Indeed, this is further demonstrated by lack of knowledge about the exact number of children who experience one or both parents imprisoned. It is estimated that around 312,000 children have a parent in an English or Welsh prison each year [5].

Mothers in prison are more likely to have been the primary caregiver compared to fathers in prison [2]. The Prison Reform Trust found that in England only 9% of children are living with their father when their mother is imprisoned, compared to almost 75% of children living with their mother when their father is imprisoned [3]. Another report estimated that 17,000 children are affected by maternal imprisonment annually [6].

Having either parent imprisoned can cause significant family disruption, instability, confusion and uncertainty for children, their parents and carers [3]. Family challenges can include physical, emotional and financial difficulties with loss of income, accommodation and moving, child-care support and stigma. There are also detrimental health implications for the child and family members. The grief of separation from parents, being unable to see or talk to their incarcerated parent can be extensive [7–9]. Some postulate that it is the process of imprisonment itself, or the environment children are living in prior to their parents imprisonment, that can augment poor health and wellbeing [10,11]. In any case, having a parent who is or has been imprisoned is considered to be an adverse childhood experience (ACE) and whilst there is debate around screening for ACEs, there is an association between suffering childhood adversity and long-term poorer outcomes, including for health and wellbeing [12].

Research specifically examining the impact of parental imprisonment on their children includes a wide variety of outcome measures from children's imprisonment rates to education, language ability and employment as well as their physical and mental health and wellbeing [13]. There is mixed evidence about the health impacts of children who experience their mother being imprisoned and their father being imprisoned [14]. A systematic review and meta-analysis focusing on post-traumatic stress disorder (PTSD) found that parental imprisonment, particularly maternal, was significantly associated with PTSD for the child, whilst another found less clear links between maternal imprisonment and mental health conditions [15,16].

Poehlmann-Tynan and Turney summarised the health implications from several longitudinal population-based studies reviewing the ages in which children experience imprisonment of their parents and the child's wellbeing [14]. Whilst their synthesis showed a negative association with paternal incarceration, there was a lack of consistency in the research findings and no robust evidence of causality. They also highlighted a greater need to understand the age-related effects and the frequency of parental involvement with the criminal justice system [14].

The wide variation in study methods, designs, and how health and wellbeing outcomes are measured makes comparison of study findings challenging. Many discuss findings relating to experiencing parental imprisonment, but do not differentiate between children experiencing their mother being imprisoned (MI), or their father being imprisoned (FI) or both parents being imprisoned. This is particularly relevant, as evidence suggests that experiencing their MI is likely to be more disruptive to the children as mothers are much more likely than fathers to be the primary caregiver. When children experience their FI, they are more likely to remain in the family home with their mother; when a mother is imprisoned her children are much more likely to move, be taken into statutory care or grandparental care'. A Home Office Research Study in 1997 found that when mothers are imprisoned, only 5% of children remain in their family home [17]. This can result in children having to change schools and disrupt their education as well as isolation and separation from friends and their community [18]. Children may not only have to move living accommodation, but they may have to travel considerable distances to visit their parent in prison, as highlighted by Hagan and Coleman [19]. There are only twelve women's prisons in England and none in Wales. This means that women will often be further from their families in comparison to men, and Prison Reform Trust noted a significant number being held more than 100 miles from home. This means visiting will

be more challenging [20]. These disruptive factors isolate the children and confound grief, which may be expressed in depression, anger and aggressive behaviour [21]. Other reviews explored the impact of maternal versus paternal imprisonment on their children by comparing results from studies with populations only exposed to MI or FI, each of which had their own study design. The variety in study methods and outcomes measured also have implications for the interpretation of findings; for example, the child's age when parental imprisonment occurs[4,15,22,16], frequency and length of parental imprisonment,[4,14,22] living and family environments[4,15,23], demographic[14,22,16] and socio-economic status[14,23,24] are all likely to have an effect on outcomes [4,14–16,22–24].

To date there has been no systematic synthesis of the existing evidence of outcomes relating to children experiencing maternal versus paternal imprisonment in studies of the same population. This scoping review therefore aims to fill this gap by reviewing the published evidence about the health impacts of maternal versus paternal imprisonment to enable a better understanding of the differential impacts on affected children and to identify where gaps in the evidence remain.

## Methods

The following scoping review was undertaken using the methodology described by Arksey and O'Malley [25]. The five stages of the scoping review process included: 1) Identifying the research question; 2) Identifying relevant studies; 3) Study selection; 4) Charting the data; 5) Collating, summarising and reporting the results [25]. The PRISMA Extension for Scoping Reviews (PRISMA-ScR) guidance was used during this review [26].

◦ *Identifying the research question*

First, the following research question was identified, 'How do the physical, mental and behavioural health, along with healthcare service use differ between children who experience their mother being imprisoned, compared to those who experience their father being imprisoned?'. We included studies that had outcomes relating to health and behavioural health of an individual who had experienced maternal or paternal imprisonment.

◦ *Identifying relevant studies*

We conducted a systematic literature search, using multiple databases including Medline, Embase, CINAHL, Cochrane, PyschINFO, Web of Science, Delphis and IBBS to reflect the breadth of potential health outcomes in June 2024 with no publication date restrictions.

To ensure that all relevant research was considered, there were no date or language restrictions. The search terms were broad to maximise sensitivity. These included: imprison* or incarcerat* or jail* or prison* or gaol*; Child* or "young person" or adolescen* or teen* or youth*; Maternal* or mother* or mum* or mom* or mam*; paternal* or father* or dad* or papa*.

Review articles were not included, but their reference lists were hand-searched. Grey literature was not included.

◦ *Study selection*

Three authors (NG, EP and AD) screened articles by title and abstract to ensure that there was data for both maternal and paternal imprisonment. Full text for potentially eligible studies were then sourced. If there was uncertainty about the relevance of the study, it was discussed between NG, AD and EP and an agreement was reached.

Studies were included if they compared anyone who as a child or adolescence experienced or at the time of the study were experiencing their mothers being imprisoned to no parental imprisonment and children experiencing their father being imprisoned (or both) and their outcomes related to child physical health, mental health, behavioural health, or healthcare service use.

Studies examining the impact of parental imprisonment due to prisoners of war or immigration removal centres or secure psychiatric units were excluded as the circumstances relating to these situations were likely to be different to our

area of interest. Studies examining the impact on family relations were not included as whilst they may impact on health, they are not health conditions.

Table 1 outlines the inclusion and exclusion criteria.

○ *Collating, summarising and reporting results*

The data was charted in a table with headings: author month and year, title, population, key findings. A selection of full texts were reviewed by NG, AD and EP.

Data was collected and grouped into common themes, including mental health, sexual health, sleep and diet. The included studies were then quality assessed independently by NG and EP, of which 25% of were discussed to ensure agreement was reached about reporting their quality.

## Quality assessment of studies

Whilst Arksey and O'Malley methodology provides a systematic approach, the papers are not assessed for their quality [27]. The NIH Quality Assessment Tool was therefore applied to assess all included studies. This toolkit provides a systematic method to evaluate studies including both Observed Cohort and Cross-Sectional studies [27,28]. There was one additional paper that reported the use of propensity scores, and for this paper we used the checklist for economic evaluation, critical appraisal tools for use in JBI Systematic Reviews [29]. The included studies were quality assessed independently by NG, AD and EP, who then discussed a sample together. Agreement about quality levels were discussed and agreement reached through discussion together where there were differences or uncertainty.

## Results

The search resulted in 9,773 articles. Once the eligibility criteria were applied, there were 140 papers identified of which 69 were duplications and removed. This resulted in 71 papers being reviewed. A sample were independently reviewed and agreement on their inclusion was reached through discussion if there were any uncertainties. One relevant paper was identified by reviewing the reference list for review papers. Ultimately, 20 studies met the inclusion criteria (Fig 1) [11,12,30–48].

Table 1. Study inclusion and exclusion criteria.

| Inclusion | | Exclusion |
|---|---|---|
| **Participants** | Any children or adolescence who experienced their mother and father being imprisoned | If parent was detained in immigration removal centres or secure psychiatric units or was a prisoner of war |
| | Male or Female | |
| **Intervention** | Experience of maternal imprisonment or father imprisonment, or both, within the same study population. The comparison group may be experiencing no parental imprisonment | Only focused on maternal or only paternal imprisonment |
| **Outcome** | Any health condition including:<br>• Physical health<br>• Mental health<br>• Behavioural health, for example behavioural challenges and sleeping.<br>The paper must report health outcomes for children who have experienced maternal imprisonment and children who have experienced paternal imprisonment | Not a health condition, such as<br>• income,<br>• homeless,<br>• school achievement<br>• educational level<br>• sporting achievement<br>• involvement in the criminal justice system<br>• family relations |
| **Study** | Prevalence studies, cross-sectional studies, cohort studies, case control studies, surveys<br>Any country, any date of publication, any language<br>Published and un-published data | Policy, opinion, review articles. (Review articles were excluded but their reference lists were used to check for any missed papers) |

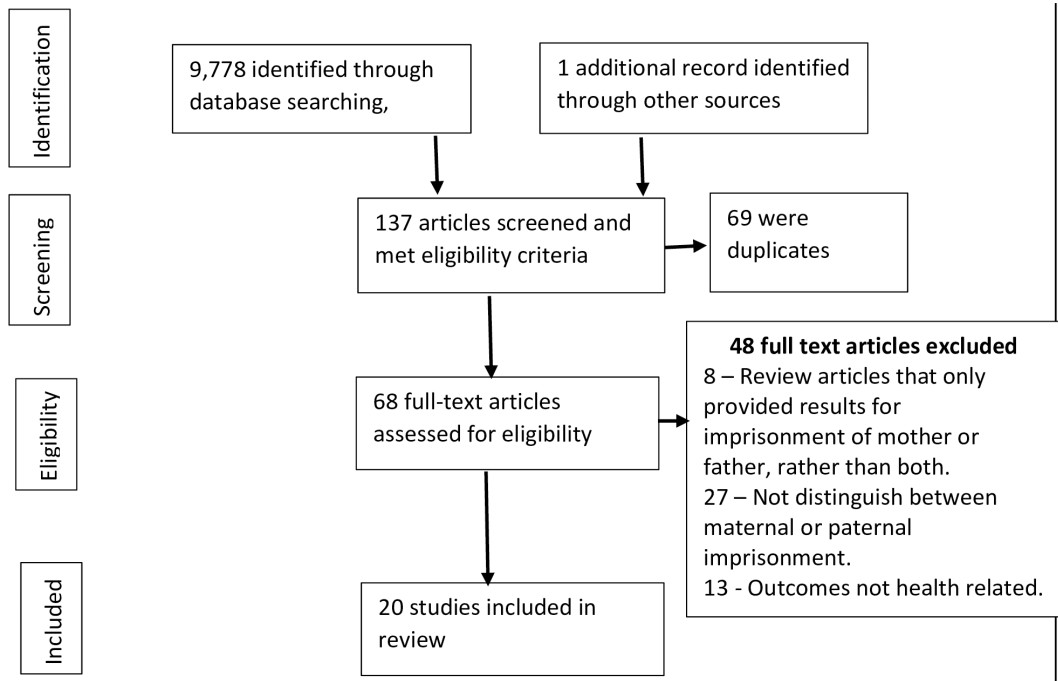

**Fig 1. The Impact of maternal versus paternal imprisonment on their children's health: A scoping review literature search.**

All 20 studies included papers that compared data relating to outcomes for children and adolescents who had experienced maternal or paternal imprisonment when they were under 18 years to children with no parental imprisonment. Three of these also compared outcomes for children experiencing maternal and paternal imprisonment to each-other [11,44,47]. This meant that the main comparator was the population who had not experienced parental imprisonment (NPI) as a child.

Whilst all the studies examined the difference between parental gender incarceration, only five of the studies analyse data about children's gender [31,36,37,46,47]. Three related to depression, one to deaths and the fifth related to c-reactive protein (CRP) and inflammation results.

Eighteen studies were based in the United States of America, with only two studies using populations in other countries; one undertaken in Denmark that reviewed child mortality data when their parents were imprisoned and the other undertaken in South Korea and reviewed parental reports of their child having depression [46,47].

There were two studies that provided data for 13 of the 20 papers included. The most frequent study data used was the National Longitudinal Study of Adolescence Health (Add Health) with eight studies using data from it [12,30–36]. The second most common data source was from the Fragile Family and Child Wellbeing Survey from which five of the publications were based [37–41]. Further details about the two study populations can be found in Appendix C.

Add Health, a longitudinal study, recruited a nationally representative sample of adolescents from United States of American in Grades 7–12 in 1994/94 and followed them until 2007/08 with four waves of surveys. 90,118 students were initially surveyed in school, with 20,745 asked to complete in-home surveys and had their weight and height taken during the face-to-face interviews. Wave 4 (number = 15,701) was undertaken in 2008–09 when in-home surveys were undertaken along with their weight, height, waist, blood spot and urine samples. Participants were asked if they had experienced one or both parent or primary care being in imprisoned [12,30–36].

The Fragile Families and Child Wellbeing Survey, also undertaken in the United States of America, was a longitudinal study which recruited parents of children born between 1998 and 2000, in random cities with populations greater than 200,000 people, in sampled hospitals. It followed both married and unmarried couples until the child was nine years of age. There was purposeful oversampling of low-income, "unwed" couples. Participants were asked if they or their partner had a history of having spent time in jail or prison on or before the child was five years of age [37–41].

Findings comprised four main categories of health: physical health, mental health, behavioural health (including substance use, sexual practice and nutrition) and healthcare service use.

### Physical health

Physical health outcomes were measured across four studies and covered cancer, high cholesterol, hypertension, diabetes, asthma; migraine, epilepsy, hepatitis, and human immunodeficiency and acquired immunodeficiency syndrome (HIV/AIDS), weight, C-Reactive Protein (CRP) levels and finally death. No two studies reviewed the same physical health outcome although one reviewed data about children being overweight, whilst another studied children being obese. All four of the studies used the comparator group being children with no experience of parental imprisonment, as opposed to comparing those who have experienced maternal imprisonment to those who have experienced paternal imprisonment [30,34,37,46]. Results are outlined in Table 2.

There was a statistically significant difference between biological children who had experience FI compared to those who had not experienced NPI for having a diagnosis of high cholesterol, asthma, migraine and HIV/AIDS [34]. In a separate study, those who experienced FI compared to NPI were also significantly more likely to have high risk of low grade inflammation demonstrated by the biochemical marker C-Reactive Protein (hs-CRP) levels of 3–10 mg/L aOR: 1.44 95%CI(1.09, 1.91) compared to NPI [30]. A Danish study found that boys who experience FI were approximately 2.26 times more likely to die before reaching the age of 20 years of age than boys who experienced NPI. The cause of death was not analysed, but results were adjusted to account for changes in mortality rates across ages [46]. These studies found no significant difference for children experiencing MI compared to NPI. Conversely, a health benefit was identified during Branigan's study which showed that children who experienced MI compared to NPI or FI compared to NPI were statistically more likely not to be obese (aOR 0.43, 95%CI 0.22 to 0.83, and aOR 0.58 95%CI 0.38 to 0.87) respectively [38].

### Mental health

Nine studies reported outcomes relating to mental health, comparing children who had experienced MI to those who had experienced NPI, as well as those who had experienced FI to those who had experienced NPI [11,12,31,34,40,44,45,47,48]. Two studies had outcomes relating to mental health and compared children who had experienced FI to those of MI (See Table 3) [44,47]. Outcomes measured included depression, suicide, interpersonal psychopathy, affective psychopathy, aggressive behaviour, having a 'mental health condition', and attending mental health counselling.

***Depression and suicide.*** Five studies compared depression outcomes for those who experienced MI to those who had experienced NPI and separately compared those who experienced FI to those who had experienced NPI [12,31,34,35,40].

Lee and McDaniel used data from the Add Health study, and both found that when comparing experiencing FI to NPI or MI to NPI, symptoms of depression in exposed groups were significantly higher [34,35]. These findings were not consistent, as demonstrated by two studies concluding that only experiencing MI compared to NPI resulted in significantly higher likelihood to have symptoms of depression [12,31]. In contrast, another study found no significant difference for MI or FI in regards to children's depression when compared to NPI [40].

**Table 2. Summary of results from studies looking at physical health outcomes for people who experience their father being imprisoned or their mother being imprisoned.**

| | Lee et al [34] | | | | Branigan [37] | | | | Wildeman [46] | | | | Boch [30] | | | |
|---|---|---|---|---|---|---|---|---|---|---|---|---|---|---|---|---|
| | MI versus NPI | | FI versus NPI | | MI versus NPI | | FI versus NPI | | MI versus NPI | | FI versus NPI | | MI versus NPI | | FI versus NPI | |
| | OR | (95%CI) | aOR | (95%CI) | aOR | (95%CI) | aOR | (95%CI) | aOR | (95%CI) | aOR | (95%CI) | aOR | (95%CI) | aOR | (95%CI) |
| Asthma (HCPD) | 1.16 | (0.77,1.75) | 1.30* | (1.05,1.62) | NR | | NR | | NR | | NR | | NR | | NR | |
| Cancer (HCPD) | 1.42 | (0.50,3.99) | 0.98 | (0.50,1.92) | NR | | NR | | NR | | NR | | NR | | NR | |
| Diabetes (HCPD) | 0.57 | (0.19,1.73) | 0.98 | (0.63,1.53) | NR | | NR | | NR | | NR | | NR | | NR | |
| Epilepsy (HCPD) | 0.79 | (0.26,2.44) | 1.43 | (0.78,2.64) | NR | | NR | | NR | | NR | | NR | | NR | |
| Heart Disease (HCPD) | 0.16 | (0.02,1.29) | 1.62 | (0.77,3.39) | NR | | NR | | NR | | NR | | NR | | NR | |
| Hepatitis (HCPD) | NR | | 0.25 | (0.05,1.36) | NR | | NR | | NR | | NR | | NR | | NR | |
| HIV/AIDS (HCPD) | 0.10 | (0.00,4.19) | 4.05* | (1.03,15.88) | NR | | NR | | NR | | NR | | NR | | NR | |
| High cholesterol (HCPD) | 1.17 | (0.64,2.15) | 1.31* | (1.01,1.69) | NR | | NR | | NR | | NR | | NR | | NR | |
| Hypertension (HCPD) | 0.95 | (0.52,1.73) | 1.22 | (0.96,1.55) | NR | | NR | | NR | | NR | | NR | | NR | |
| Migraine (HCPD) | 1.08 | (0.66,1.79) | 1.26* | (1.03,1.54) | NR | | NR | | NR | | NR | | NR | | NR | |
| Obesity (Measurement) | 0.95 | (0.70,1.31) | 0.94 | (0.81,1.10) | NR | | NR | | NR | | NR | | NR | | NR | |
| Overweight (Measurement) | NR | | NR | | **0.43** | **(0.22,0.83)** | **0.58** | **(0.38,0.87)** | NR | | NR | | NR | | NR | |
| Death (HCPD) | NR | | NR | | NR | | NR | | Female child 1.25<br>Male child 2.22 | (0.17,9.12)<br>(0.86,5.75) | Female child 0.78<br>Male child **2.26\*\*** | (0.28,2.23)<br>**(1.35,3.82)** | NR | | NR | |
| Hs-CRP >10 mg/L (blood measurement) | NR | | NR | | NR | | NR | | NR | | NR | | Female child 0.76<br>Male child 1.03 | (0.46, 1.27)<br>(0.45, 2.33) | Female child 1.23<br>Male child 1.01 | (0.89, 1.72)<br>(0.58, 1.75) |
| Hs-CRP 3–10 mg/L (blood measurement) | NR | | NR | | NR | | NR | | NR | | **NR** | | Female child 0.97<br>Male child 1.06 | (0.61, 1.55)<br>(0.58, 1.95) | Female child **1.44**<br>Male child 1.01 | **(1.09, 1.91)**<br>(0.71, 1.43) |

MI – Maternal imprisonment, FI – father imprisonment, NPI – no parental imprisonment.

(HCPD) – health care professional diagnosis. SR – child self-reporting or parents self-reporting about their child.

aOR (95%CI) – adjusted odds ratio, 95% confidence intervals.

**Bold – statistically significant** (where available * means $p<0.05$, ** means $p<0.0 1$ *** means $p<0.001$) NR – not reported.

Quinn's study reviewed outcomes relating to planning suicide in the previous 12 months and found those who experienced MI compared to those who experienced NPI had significantly higher rates but those who experienced FI did not [45].

***Post-traumatic stress disorder and anxiety***. Only one study reported outcomes relating to PTSD or anxiety; this showed that those who experienced FI were statistically more likely to have anxiety or PTSD compared to those with NPI but those who experienced MI were not [34].

**Psychopathy and aggressive behaviour**. Psychopathy and aggressive behaviour outcomes were analysed in two separate studies. Children who experienced FI compared to those who experience NPI were more likely to experience interpersonal psychopathy [11]. The other study, undertaken by Fritsch et al found that children who experienced FI were more likely to be internalising their feelings [48]. In contrast to this, those who experienced MI were more likely to experience affective psychopathy compared to those who experience NPI [11]. Fritsch et al found that children experiencing MI were more likely to be acting-out and being aggressive.

**Comparison of mental health outcomes between children who have experienced MI and FI**. Three studies compared outcomes for children who experienced MI compared to those who experienced FI. These studies reviewed two different mental health outcomes. All used FI as the reference group [11,44,47].

Tasca found that MI rather than FI were significantly more likely to report the child having a mental health condition [44]. Woo et al also found parents were more likely report their child having symptoms of depression if the MI rather than FI [47]. Thomson found that MI rather than FI resulted in greater reporting of experiencing affective psychopathy. Contrastingly, those who experienced MI were significantly less likely to experience interpersonal psychopathy [11]. In contrast, a separate study found no significant difference for MI or FI in regards to children's depression [40].

***Mental health counselling***. Children who experienced FI had significantly greater odds of using mental health counselling that those with NPI.

## Behavioural health

This section covers behavioural health outcomes and includes findings relating to: substance, alcohol and gambling use; sexual practice and relationships (including sexually transmitted infections) and health behaviour (including diet and sleep).

***Substance and alcohol use***. Three papers examined outcomes relating to substance and alcohol, all used the comparator of NPI and all looked at "problem" drug or substance use [12,31,32]. All three studies found a statistically significant difference between comparing outcomes for those who have experienced FI compared to NPI and 'problem' drug or substance use (Table 4). Whilst two studies found significant differences when comparing those who have experienced MI to NPI, Heard-Garris's findings were non-significant [12,31,32].

***Diet***. Outcomes relating to diet were reviewed by two studies as shown in Table 5 [32,41]. Jackson et al identified that there was significantly higher consumption of sugary drinks, salty snacks, starch consumption and sweet consumption when comparing those exposed to FI to NPI and those exposed to MI to NPI [41].

Whilst in a separate study, Heard-Garris also reported findings relating to consumption of sugary drink/soda and found consistent findings for those exposed to FI compared to NPI, there were no significant findings for those exposure to MI compared to NPI [32].

***Sleep***. Jackson. also studied sleep data and identified that when comparing children experiencing FI to NPI and children experiencing MI to NPI, both had significant higher levels of short sleep (duration under eight hours) [41]. This was in contrast to the study undertaken by Branigan et al where no significant difference was found for either group [37].

***Sexual health***. Sexual health outcomes were measured in four studies, which all used NPI as the comparator (Table 6) [32,33,36,43]. Two papers looked at early age sexual initiation but showed contradictory findings [33,43]. One study found that those who experienced MI compared to NPI were significantly more likely to have 'early sex' (under 15 years of

**Table 3. Summary of results from studies looking at mental health outcomes for people who experienced their farther being imprisoned or their mother being imprisoned or both.**

| | Lee study [34] (HCPD) | | | | McDaniel [35] (SR) | | | | Foster [31] (CES -D scale) | | | Kopak [12] (CES -D scale) | |
|---|---|---|---|---|---|---|---|---|---|---|---|---|---|
| | MI v NPI | | FI v NPI | | MI v NPI | | FI v NPI | | MI v NPI | | FI v NPI | MI v NPI | |
| | aOR | 95% CI | aOR | 95% CI | SE | P value | SE | P value | aOR | 95% CI | aOR 95% CI | SE | P value |
| Anxiety | 1.47 | 0.88, 2.46 | **1.51** | **1.23, 1.85** | NR | | NR | | NR | | NR | NR | |
| Aggressive | NR | | NR | | NR | | NR | | NR | | NR | NR | |
| Depression | **1.6** | **1.02, 1.80** | 1.43 | **1.15,1.78** | 0.1 | 0.04 | 0.05 | 0.02 | 2.21 | 1.31, 3.74 | NS | 0.85 | 0.38 |
| PTSD | 1.48 | 0.66, 3.35 | **1.72** | **1.21, 2.45** | NR | | NR | | NR | | NR | NR | |
| MH Counselling | NR | | NR | | NR | | NR | | NR | | NR | NR | |
| Parent reported MH condition for their child | NR | | NR | | NR | | NR | | NR | | NR | NR | |
| Planned suicide in previous 12 months | NR | | NR | | NR | | NR | | NR | | NR | NR | |

MI – Maternal imprisonment, FI – father imprisonment, NPI – no parental imprisonment v – versus.

OR odds ratio, aOR (95%CI) – adjusted odds ratio, 95% confidence intervals.

**Bold – statistically significant,** NS not significant, NR – not reported.

HCPD – health care professional diagnosis, SR – child self-reporting or parents self-reporting about their child, CES-D – Centre of Epidemiological Studies – Depression scale, CBC – Child Behavioural Checklist.

SE- standard error.

| Geller [40] (CBC – scale) | | | Heard-Garris [32] (SR) | | | | Quinn [45] (SR by parents) | | | | Woo [47] (SR by parents) | Tasca [44] (SR by parents) |
|---|---|---|---|---|---|---|---|---|---|---|---|---|
| FI v NPI | MI v NPI | FI v NPI | MI v NPI | | FI v NPI | | MI v NPI | | FI v NPI | | MI v FI (FI comparator) | MI v FI (FI comparator) |
| SE, p value | aOR 95% CI | aOR 95% CI | aOR | 95% CI | aOR | 95% CI | aOR | 95% CI | aOR | 95% CI | P value | P value |
| NR | NR | NR | NR | | NR | | NR | | NR | | NR | NR |
| NR | NS | NS | NR | | NR | | NR | | NR | | NR | NR |
| NS | NS | NS | NR | | NR | | NR | | NR | | < 0.01 | NR |
| NR | NR | NR | NR | | NR | | NR | | NR | | NR | NR |
| NR | NR | NR | 1.14 | 0.71, 1.85 | 1.6 | 1.2, 2.08 | NR | | NR | | NR | NR |
| NR | NR | NR | NR | | NR | | NR | | NR | | NR | p<0.05 |
| NR | NR | NR | NR | | NR | | 15.1 | 0.01, 0.37 | 0.74 | 0.22, 2.49 | NR | NR |

**Table 4. Summary of results from studies looking at substance and alcohol use outcomes for people experience their farther being imprisoned or their mother being imprisoned.**

| | Heard-Garris study [32] SR | | | | Foster study [31] SR | | | | Kopak study [12] SR | | | |
|---|---|---|---|---|---|---|---|---|---|---|---|---|
| | MI versus NPI | | FI versus NPI | | MI versus NPI | | FI versus NPI | | MI versus NPI | | FI versus NPI | |
| | aOR | 95% CI | aOR | 95% CI | aOR | 95% CI | aOR | 95% CI | aOR | SE | aOR | SE |
| Cigarette smoking (within 30 d) | **1.77***** | **1.26, 2.50** | **1.53***** | **1.28, 1.82** | NR | | NR | | NR | | NR | |
| Illicit IV drug | 0.96 | 0.19, 4.77 | **2.54**** | **1.27, 5.12** | NR | | NR | | NR | | NR | |
| Lifetime cocaine use | NR | | NR | | NR | | NR | | 2.36 | 0.65** | 1.95 | 0.31** |
| Lifetime drug use other than marijuana or cocaine | NR | | NR | | NR | | NR | | 1.98 | 0.48** | 1.46 | 0.18** |
| Lifetime marijuana use | NR | | NR | | NR | | NR | | 1.90 | 0.31** | 1.56 | 0.13** |
| Prescription drug inappropriate use | **1.61*** | **1.02, 2.55** | **1.46***** | **1.20, 1.79** | NR | | NR | | NR | | NR | |
| "Problem" drinking | **1.70**** | **1.17, 2.49** | **1.46***** | **1.21, 1.75** | NR | | NR | | NR | | NR | |
| "Problem drug"/ substance use | 1.38 | 0.59, 3.23 | **1.74***** | **1.25, 2.41** | 2.09* | 1.10, 3.97 | **1.74**** | **1.15, 2.64** | 0.42 | 0.15** | 0.17 | 0.07* |
| Gambling | 1.34 | 0.56, 3.20 | 1.77 | 0.98, 3.20 | NR | | NR | | NR | | NR | |

MI – Maternal imprisonment, FI – father imprisonment, NPI – no parental imprisonment, IV-Intra-venous.

aOR (95%CI) – adjusted odds ratio, 95% confidence intervals.

**Bold – statistically significant** (where available * means $p < 0.05$, ** means $p < 0.0 1$ *** means $p < 0.001$).

NR – not reported.

SR – child self-reporting or parents self-reporting about their child.

SE – standard error.

age), whereas there was no significant finding if for FI compared to NPI [33]. However, another, using different data from USA, found the opposite and those who experienced FI compared to NPI was significantly more likely to result in having early sex (under 13 years of age), whereas there was no significant finding if experiencing MI compared to NPI [43].

Le G et al also found that experiencing MI increased the odds of increased sexually transmitted infections in comparison to NPI [43]. Those experiencing FI were statistically significantly more likely to have sexually transmitted infections. This contrasted with Roettger who reviewed a different outcome of lifetime risk of sexually transmitted infection (STI). Roettger, using Add Health data, found that MI was not significantly associated with children's higher lifetime risk of an STI after adjusting for confounders, but experiencing FI of their biological father was significant, and were more likely to have had an STI in the previous 12 months [36].

### Healthcare use, health behaviour and self-reported health

***Subjective health and healthcare use.*** Two studies analysed data relating to healthcare use and general health status (Table 7) [32,34].

Self-reported poor health was evaluated by Lee et al. [34]. The odds of individuals who had experienced FI reported having fair or poor health significantly more than those with NPI [34]. This significant difference was not found for those who experience MI.

Table 5. Summary of results from studies looking at general health, health consultations and diet for people experience their farther being imprisoned or their mother being imprisoned.

| | Heard-Garris [32] SR | | | | Branigan [37] SR | | | | Jackson study [41] SR | | | |
|---|---|---|---|---|---|---|---|---|---|---|---|---|
| | MI versus NPI | | FI versus NPI | | MI versus NPI | | FI versus NPI | | FI versus NPI | | MI versus NPI | |
| | aOR | 95% CI | aOR | 95% CI | aOR | 95% CI | aOR | 95% CI | aOR | 95% CI | aOR | 95% CI |
| Fast food | 1.10 | 0.69, 1.75 | 1.15 | 0.94, 1.43 | NR | | NR | | **2.09** | **1.32, 3.30** | 1.02 | 0.76, 1.37 |
| Salty snack | NR | | NR | | NR | | NR | | **1.67** | **1.14, 2.46** | **1.41** | **1.11, 1.79** |
| Starch consumption | NR | | NR | | NR | | NR | | 1.26 | 0.84, 1.87 | **1.41** | **1.12, 1.77** |
| Sugary drinks/soda | 1.29 | 0.90, 1.87 | **1.37\*\*** | **1.10,1.69** | NR | | NR | | **1.31** | **1.04, 1.66** | **1.80** | **1.23, 2.64** |
| Sweets consumption | NR | | NR | | NR | | NR | | **2.05** | **1.38, 3.06** | **1.50** | **1.16, 1.95** |
| Fitness center use ≥ 4*a wk | 0.71 | 0.39, 1.31 | 0.75 | 0.54,1.04 | NR | | NR | | NR | | NR | |
| Consistent bedtime | NR | | NR | | 1.21 | 0.74,1.97 | **0.83** | **0.70, 0.98** | NR | | NR | |
| Short sleep duration (<9 hours) | NR | | NR | | 1.20 | 0.71, 2.02 | 1.20 | 1.00, 1.44 | **1.49** | **1.04, 2.14** | **1.28** | **1.04, 1.57** |
| Sleep problems | NR | | NR | | NR | | NR | | 1.16 | 0.81, 1.66 | **1.47** | **1.20, 1.80** |
| Total sleep duration | NR | | NR | | 0.00 | −0.25, 0.25 | −0.08 | −0.17, 0.01 | NR | | NR | |

MI – Maternal imprisonment, FI – father imprisonment, NPI – no parental imprisonment.

aOR (95%CI) – adjusted odds ratio, 95% confidence intervals.

**Bold – statistically significant** (where available * means p < 0.05, ** means p < 0.0 1 *** means p < 0.001).

NR – not recorded.

SR – child self-reporting or parents self-reporting about their child.

Heard-Garris compared healthcare use outcomes and identified children who experienced MI or FI had significantly greater levels of worsening health problems and were significantly more likely to have foregone healthcare (should have had medical care but did not attend), when compared to experiencing NPI [32].

Children who experienced MI compared to NPI were also significantly more likely to access usual healthcare in emergency department or non-primary care settings and not attend annual dental examination. These were not significant when comparing children who experienced FI to NPI [32]

## Quality assessment of studies

Overall the papers were of mixed quality, as assessed using the National Institutes of Health (NIH) Quality Assessment Tool. The good papers recognised the gaps in their studies and discussed their limitations. Ten of the papers were of good quality and one paper that was poor/fair. It was not possible to state if any of the outcome assessors were blinded to the exposure status of participants. It was unclear in most papers if the participation rate of eligible persons was at least 50% or if the loss to follow-up after baseline was 20% or less. Five papers did not consider, measure or adjust for key confounders. Within the studies that did adjust for confounding, the factors adjusted for varied even when reviewing outcomes for the same study population, e.g., Add Health papers. Further explanation about study quality is found in Appendix D.

## Discussion

This scoping review found that exposure to parental imprisonment, be it maternal or paternal, has wide ranging health implications for children from physical to mental and behavioural health. Whilst the literature demonstrated the negative impacts that can occur if parents are imprisoned, there are still substantial gaps in the evidence, particularly for those children living in the United Kingdom. Only 20 studies reported child outcomes relating to physical health, mental health, wellbeing or healthcare service use, separated into if the children are exposed to MI compared to NPI and FI compared

**Table 6. Summary of results from studies looking at sexual health outcomes for people experience their farther being imprisoned or their mother being imprisoned.**

| | Heard-Garris study [32] (SR) | | | | Le G study [33] (SR) | | | | Roettger study [36] (HCPR) | | | | Nebbitt study [43] (SR) | | |
|---|---|---|---|---|---|---|---|---|---|---|---|---|---|---|---|
| | MI v NPI | | FI v NPI aOR (95%CI) | | MI v NPI aOR (95%CI) | | FI v NPI aOR (95%CI) | | MI v NPI aOR (95%CI) | | FI v NPI aOR (95%CI) | | MI v NPI | FI v NPI | |
| | aOR | 95%CI | aOR | 95%CI | aOR | 95%CI | aOR | 95%CI | aOR | 95%CI | aOR | 95%CI | OR P value | OR | P value |
| 10 + lifetime sexual partners | **1.55\*** | **1.08, 2.22** | **1.19\*** | **1.01, 1.41** | NR | | NR | | NR | | NR | | NR | NR | |
| Sex in exchange for money | **2.16\*** | **1.05, 4.45** | 1.03 | 0.57, 1.85 | NR | | NR | | NR | | NR | | NR | NR | |
| Early sex initiation | NR | | NR | | **3.6 \*\*\*** | **1.9, 6.7** | 1.3 (.9, 1.9) | | NR | | NR | | <u>NS</u> | **10.5** | **<0.05** |
| Inconsistent condom use | NR | | NR | | **3.4 \*** | **1.3, 8.0** | 1.0 (.8, 1.3) | | NR | | NR | | NR | NR | |
| Positive STI test | NR | | NR | | **5.5 \*\*** | **1.7, 17.6** | **1.7 \*** | **1.1, 2.8** | NR | | NR | | NR | NR | |
| Positive STI in 12 months prior to interview | NR | | NR | | NR | | NR | | 1.24 | 0.81, 1.86 | **1.19** | **1.01, 1.41** | NR | NR | |
| Lifetime risk of STI | NR | | NR | | NR | | NR | | 1.20 | 0.90, 1.61 | **1.22** | **1.09,1.37** | NR | NR | |

MI – Maternal imprisonment, FI – father imprisonment, NPI – no parental imprisonment, STI-sexually transmitted infection.

aOR (95%CI) – adjusted odds ratio, 95% confidence intervals, OR- odds ratio.

**Bold – statistically significant** (where available * means p < 0.05, ** means p < 0.0 1 *** means p < 0.001) NS – not significant.

**NR – not reported.**

**SR –** child self-reporting or parents self-reporting about their child, **HCPR – health care professional reported.**

**Table 7. Summary of results from studies looking at general health, health consultations and diet outcomes for people experience their farther being imprisoned or their mother being imprisoned.**

| | Heard-Garris [32] SR | | | | Lee study [34] SR | | | |
|---|---|---|---|---|---|---|---|---|
| | MI versus NPI | | FI versus NPI | | MI versus NPI | | FI versus NPI | |
| | aOR | 95%CI | aOR | 95%CI | aOR | 95%CI | aOR | 95%CI |
| Annual dental exam | **0.67\*\*** | **0.50, 0.90** | 0.89 | 0.75, 1.07 | NR | | NR | |
| Foregone healthcare | **1.65\*\*** | **1.20, 2.27** | **1.22\*** | **1.02, 1.47** | NR | | NR | |
| Poor Health | NR | | NR | | 0.97 | 0.53, 1.79 | **1.28\*** | **1.03, 1.59** |
| Usual source of care in ED or non-primary care | **2.36\*\*\*** | **1.51, 3.68** | 1.22 | 0.98, 1.53 | NR | | NR | |
| Worsening health problems | **1.29** | **0.76, 2.22** | **1.51\*\*** | **1.18, 1.95** | NR | | NR | |

MI – Maternal imprisonment, FI – father imprisonment, NPI – no parental imprisonment.

SR – child self-reporting or parents self-reporting about their child.

aOR (95%CI) – adjusted odds ratio, 95% confidence intervals.

**Bold – statistically significant** (where available * means p < 0.05, ** means p < 0.0 1 *** means p < 0.001).

NR – not recorded.

to NPI, and only three directly compared outcomes for those experiencing MI to those experiencing FI. This sparsity of literature resulted in challenges addressing the original study question about how health and wellbeing outcomes differ for children experiencing maternal versus paternal imprisonment and no clear conclusions can be drawn [11,44,47]. However, the findings suggested there is evidence that experiencing parental imprisonment, be it MI or FI, will impact the child's health and wellbeing. This is consistent with findings from the broader literature that demonstrates experiencing any parental incarceration will impact children's health and well-being [4,16,33,36].

The impact is mostly thought to be detrimental but on occasion, such as in child abuse, there can be some relief for the child [48,49]. However, devastating implications can occur due to the loss of a parent, uncertainty about future and how to visit the imprisoned parent, loss of stability and a care provider, disruption to family relations, as well as economic loss [48,49]. Children may also be affected by external stigma and bullying and be fearful of others finding out about their parent/s being imprisoned so they become be socially isolated [36,48,49]. These factors in turn, can impact negatively on the health and well-being of children, especially their mental health.

This scoping review highlights how inconsistent the data collection, study design and findings are for the literature that reviewed the impact maternal and paternal imprisonment had on children. It also demonstrates that there is a deficit within the literature to examine the potential poor health outcomes related to experiencing a mother or a father who provides care, being incarcerated compared to outcomes for children who experience NPI. However, this review provided the first synthesis of existing literature relating to maternal and paternal imprisonment to gain a holistic understanding of the parental-gender impacts of incarceration on their child, both physically and psychologically.

Comparisons of studies was negatively impacted as a consequence of methodological limitations. These included variation in study design, lack of consistent population demographics and inconsistent outcome measurement. Consequently, these factors preclude meaningful interpretation of the studies available.

Murray and Farrington concluded that whilst there are limited studies relating to child health, MI, along with experiencing longer periods of imprisonment time or more punitive social contexts can worsen health outcomes [4]. Minson similarly found that material imprisonment is associated with children experiencing confusion and resentment from families members they move to. They also experience confounding grief as they not only experience emotions relating to grief including missing their mother and a lack of contact, but it becomes confounding as their grief is hidden and unacknowledged due to the stigma of their mother being in prison and a societal belief that they deserve to be there [18]. Children may require local authority placements if family and friends are unable to provide care for them which can be multiple and being part of the care system is itself linked with having poorer mental and physical health outcomes [18,50].

Two separate reviews by Wildeman et al and Poehlmann-Tyann et al concluded that whilst studies investigating the impact of experiencing FI was detrimental to children's health studies relating to MI were less clear [14,16]. This could be due to far fewer women are imprisoned, resulting in smaller study populations and meaning studies are underpowered and less likely to be able to detect a difference. Another hypothesis could relate to the findings by Heard-Garris, that concluded children experiencing MI were more likely to attend non-primary care settings to seek medical attention than those experiencing NPI. This was not significant in the group experiencing MI compared to NPI. By using non-primary care settings to seek medical care, conditions may be missed [32].

There were further gaps in the published evidence which included consideration about how health outcomes for children experiencing MI or FI were influenced by their relationship and interaction prior to being imprisoned, nor the level of contact whilst their parent was imprisoned and how that might impact on the child's health. Studies also did not identify if the parent imprisoned was the primary caregiver, what interim care was required and provided to the child. If there were other household members who were a parent to the child, or family or friends who could provide for the child, otherwise the need for foster and social care. These changes could potentially result in a lack of consistent boundaries or progress monitoring which further negatively affects health and wellbeing. Finally, none of the studies considered the influence that the type of offence committed by the child's parent could have on children's health and how it might as a

confounding factor. These could impact on their health and wellbeing, for example if they needed to move schools or cities and establish new friendships or stigma if their parents were named by local or national press. This could lead to externalising behaviours as children move to new environments and establish their own identity or be seen to be accepted by their peers. An example of this could relate to the earlier onset of sexual intercourse and sexual behaviour due to a lack of understanding about consent and hypersexuality as well as a lack of boundary setting from parents.

There is therefore a need for well conducted larger studies to disentangle the effects of imprisonment with those of likely adverse socioeconomic conditions.

During this review, it was evident that study populations vary considerably. With many of the published studies being taken from one of two American studies, this limits cohort variation and may result in repeat assessment [12,30–41]. Furthermore, the impact of factors such as age or frequency of exposure to parental imprisonment were difficult to differentiate and understanding of these elements is important. The attachment and relationships that children have with their parents prior to their imprisonment, were thought be influential on their health and behaviour.

It is evident that we have a very limited understanding about the mechanism by which parental imprisonment is implicated in poorer mental health, physical and behavioural health. This is despite the substantial implications parental imprisonment has on these individuals and the symbiotic harms they experience. Relatively little published research relates to the UK context, and furthermore, twelve of the twenty studies identified are reliant on data collected from only two American studies [12,30–41]. This results in repeat assessment of the same cohorts and further demonstrates the highly limited understanding we have for this seldom heard, underserved population [4,15,51]. This is further exacerbated by the lack of data and ability to acknowledge these children who experience symbiotic harms as a consequence of their parents being imprisoned. Although speculative, one interpretation is that as consequence of the lack of consistency, health and behaviour outcomes relating to factors can be challenging to analyse. These include taking into account the impact of experiencing as a child, the frequency of exposure to parental imprisonment, attachment and relationships prior to parental imprisonment, as well as gender of the imprisoned parent. Further research is therefore vital to ensure this population is identified, recognised and supported appropriately.

**Strengths and weaknesses of the scoping review.** This is the first scoping review to examine health outcomes for children who experience their mothers or father imprisoned. It used robust methods to ensure that all relevant papers were retrieved including a thorough search with no date or language restrictions. A unique element of this review is that studies are only included if study populations included those who experience MI and FI, rather than only cohorts that have outcomes for children who experience FI or cohorts that only have outcomes for children who experience MI. Studies were quality assessed by two independent reviewers and it was clear that there are many confounders that can be identified, but not all were measured or taken into account during the analysis and might account for some of the variations between different study findings.

The findings from this scoping review only came from a limited number of studies, and as fewer women are imprisoned than men, fewer children will experience MI than FI. The limited number of studies, as well as the sparsity of data within the limited studies, may lead to findings being underpowered and less likely to find less consistent findings or significant findings when one actually exists. The wide confidence intervals in many studies demonstrate this.

It was not possible to infer causality from these studies due to their design, sample populations and variety of confounder adjusted or not adjusted for, but they do indicate that further investigation is required to ensure that the volume and quality of evidence around this outcome is increased. This is particularly relevant in view of the increasing numbers of people incarcerated, which has risen since collection of data used in many studies.

The variability in design and consideration of confounders and effect modifiers make it difficult to compare the health implications for children when they experience MI compared to FI, or indeed any parental imprisonment. It also makes it difficult make a clear comparison between the implications of maternal compared to paternal imprisonment studies.

An example of design variation is the method used to collect data. Some studies collected data by asking children, others used data collected by asking parents about their child's health and behaviour and Wilderman used nationally linked data. Stewart-Brown showed that the method of data collection itself has an important impact on findings, particularly when undertaking research on preventative interventions for children [52]. Not using children-centred data collection methods may potentially exacerbate stress and increase the difference between study findings and everyday realities.

In some studies, it is also unclear if the MI or FI were primary caregivers or the only family for the children. None of the studies commented on consideration for same sex parenting although LGBTQI+ individuals are over-represented in prison populations. [53]Nor did any study state whether the children were the parent's biological child or not. Assumptions therefore had been made about the remaining support available for children who experience either MI or FI which may be incorrect. The available evidence does suggest that women are more likely to be the primary caregiver, and if children experience MI, they are more likely to be taken into care than children who experience FI.

Study comparison was further challenging as there was little consistency in measuring outcomes, including both which were included and how they were measured. The most frequently published outcomes in the included studies related to mental health, and findings from this scoping review mostly support previous outcome findings about the implications of parental imprisonment on children and their mental health.

Inconsistency in reporting outcome measures was evident when reviewing the age of early onset of sexual intercourse. Le G et al classified early onset of sex as under the age of 15 years but Nebbit et al classified it as under the age of 13 years [32,33]. One explanation about the variation in study findings could relate to studies adjusting (or not) for confounding factors. Confounding factors that were often considered include socio-demographics and age at analysis. However, socio-demographics is broad and can include sex, race, social class and economic impact or parents' gender.

Given there was only one data set for many of the physical health outcomes, it was not possible to establish comparative conclusions from the studies included in the scoping review. Similarly, as there was only one measurement for healthcare services use, conclusions form this study could not be established. Additionally, as these studies were undertaken in US, the healthcare systems are fundamentally different to other countries and this may influence use.

There needs to be clear consideration about the impact of any parental imprisonment has on wider families. Information about numbers of children, displacement effects that each MI and FI has on children is vital to understand the impact. Further research is needed to understand their health and social care needs to ensure that these children are adequately and appropriately supported.

The limited number of studies found for this population demonstrates that this is an invisible group of forgotten victims of the system. It is important to consider that the absence of clear significant findings, does not negate the great health needs for this cohort. This is particularly relevant when considering The United Nations Convention on the Rights of Children, which states that "States Parties shall take all appropriate measures to ensure that the child is protected against all forms of discrimination or punishment on the basis of the status, activities, expressed opinions, or beliefs of the child's parents, legal guardians, or family members" [54].

It is important that the effects of experiencing parental imprisonment are considered for children and young people. By considering alternative punishments or effectively reducing the separation between the main caregiver and their children, it may alleviate the child's suffering experienced as a result of parental punishment.

Despite the unclear findings about implications relating to health, it is well recognised that children who experience MI are vulnerable and may well need physical, mental, and emotional support. There is an urgent need to develop effective interventions to ensure appropriate support for children and families, as well as the individuals who are imprisoned.

There is a lack of robust evidence on whether the impact of maternal versus paternal imprisonment on their children differs although it is clear that children who have had a parent imprisoned are at increased risk of poor health when compared to other children. Indeed, even the most basic data on the scale of children affected is little understood and estimated at best. The disruption to children's life, accommodation, finances and support are likely to impact on their health

and further research, with careful consideration of the children of those incarcerated is vital. This should particularly relate to the comparison between experiencing maternal and paternal imprisonment and how the subsequent detrimental effects can be acknowledged and minimised. Further research is necessary to understand fully the potential similarities and differences based on parental sex, and the influence this has on their child's health and wider wellbeing. However, any further research in this area needs to question the utility and morality of the use of imprisonment for individuals who have committed crimes and who have dependent children and should also scrutinise alternatives. We need more robust evidence and a more nuanced understanding of the costs and benefits to individuals, their children, family and wider society of imprisonment.

## Supporting information

**S1 File. Appendices.** (DOCX) - Appendix A -Search terminology and papers identifiedAppendix B - Table of Papers included and Table of Papers not included due to being reviews, however, their references were checked to ensure all relevant papers were includedAppendix C - Explanation about the study populations for National Longitudinal Study of Adolescence Health (Add Health) and Fragile Families and Child Wellbeing Survey which were the two main studiesAppendix D - Quality Assessment of the studies included, using NIH Quality Assessment ToolAppendix E - Funding assoicate with scoping review

**S2 File. Data availability statement.** (DOCX)
All data was avaliable through published journals

## Author contributions

**Conceptualization:** Naomi Gadian, Emma Plugge.

**Data curation:** Naomi Gadian, Abigail Dunn, Emma Plugge.

**Formal analysis:** Naomi Gadian, Abigail Dunn, Emma Plugge.

**Investigation:** Emma Plugge.

**Methodology:** Naomi Gadian, Emma Plugge.

**Supervision:** Emma Plugge.

**Writing – original draft:** Naomi Gadian.

**Writing – review & editing:** Abigail Dunn, Donna Arrondelle, Sara Morgan, Paula Harriott, Lucy Wainwright, Emma Plugge.

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
