## [Decision Letter · Decision Letter 0]

Dear Dr. Gadian, 

Thank you for submitting your manuscript to PLOS ONE. After careful consideration, we feel that it has merit but does not fully meet PLOS ONE’s publication criteria as it currently stands. Therefore, we invite you to submit a revised version of the manuscript that addresses the points raised during the review process.

We look forward to receiving your revised manuscript.

Kind regards,

Ayse Ulgen, PhD, MGM

Academic Editor

PLOS ONE

Journal Requirements:

Reviewers' comments:

Reviewer's Responses to Questions

**Comments to the Author**

1. Is the manuscript technically sound, and do the data support the conclusions?

Reviewer #1: Partly

Reviewer #2: Yes

2. Has the statistical analysis been performed appropriately and rigorously?

Reviewer #1: Yes

Reviewer #2: N/A

3. Have the authors made all data underlying the findings in their manuscript fully available?

Reviewer #1: Yes

Reviewer #2: Yes

4. Is the manuscript presented in an intelligible fashion and written in standard English?

Reviewer #1: Yes

Reviewer #2: Yes

Reviewer #1: This is an important topic that is worthy of investigation. While the author has done well to synthesise a confusing and methodologically inconsistent body of literature, there are a number of important grammatical, punctuation, referencing, sentence structure and general content issues that require attention. The content however, is sound. I have attached a number of comments which I hope will help in the resubmission of this article. Well done authors.

Reviewer #2: Thank you for the opportunity to review this timely, useful, and well-written review. I have a few tiny comments.

Abstract is clear and succinct.

Typo in abstract “prion”

Intro

Page 2- “ACEs” not “ACE’s”

“offspring” is an odd choice of words. Can you use “children”? Can you comment on whether it matters that children are biological or not?

Methods

Research question- I suggest removing “well being.” Later you clarify you only mean health conditions, not things like poverty, so it would be clearer to just say health.

Identifying studies- please include the search terms.

Study selection- You introduce quality assessment on page 4 but only explain it on page 6. I suggest removing the mention of quality assessment until that subsection on page 6. Quality assessment is not strictly necessary for scoping reviews.

Page 7- is “unwed” the term used in the survey itself? An anachronistic term.

Page 10- that that typo

Discussion

The discussion is meandering and includes some information better placed in the intro (e.g. challenges with visits). I recommend paring it down and focusing on health outcomes, as these are all your review included (e.g. is “confusion” or “resentment” a health outcome?).

Page 20- Isn’t it Nebbit et al. who define early sex as under 13?

Line 423- parents’ needs an apostrophe

Line 429-431: remove, not part of your question

Line 460 ‘gender of parental imprisonment”- do you mean gender of imprisoned parent?

Line 488: “None of the studies commented on consideration for same sex parenting.”- thank you for acknowledging this, and you could add that the LGTBQ+ community is hyper-policed and hyper-incarcerated, and how this may therefore be particularly relevant for the children of incarcerated people.

**Do you want your identity to be public for this peer review?** For information about this choice, including consent withdrawal, please see our Privacy Policy

Reviewer #1: No

Reviewer #2: No

---

## [Author Response · Author response to Decision Letter 1]

26 Jun 2025

Dear Reviewers,

We are grateful to the reviewers for their thoughtful comments. We included the comments into one word document attached. We have used this document to methodically describe how we have addressed each of the issues they raise. Our responses are written in red.

We have ensured that grammatical, punctuation, referencing, sentence structure and content has been revised as addressed in the reviewers comments.

Thank you for your time reviewing this submission

Yours Sincerely

Naomi

---

## [Decision Letter · Decision Letter 1]

The impact of maternal versus paternal imprisonment on their children’s health: A scoping review

PONE-D-24-38282R1

Dear Dr. Gadian,

We’re pleased to inform you that your manuscript has been judged scientifically suitable for publication and will be formally accepted for publication once it meets all outstanding technical requirements.

Kind regards,

Ayse Ulgen, PhD, MGM

Academic Editor

PLOS ONE

Reviewers' comments:

Reviewer's Responses to Questions

**Comments to the Author**

Reviewer #1: All comments have been addressed

Reviewer #2: All comments have been addressed

2. Is the manuscript technically sound, and do the data support the conclusions?

Reviewer #1: Yes

Reviewer #2: Yes

3. Has the statistical analysis been performed appropriately and rigorously?

Reviewer #1: N/A

Reviewer #2: N/A

4. Have the authors made all data underlying the findings in their manuscript fully available?

Reviewer #1: Yes

Reviewer #2: Yes

5. Is the manuscript presented in an intelligible fashion and written in standard English?

Reviewer #1: Yes

Reviewer #2: Yes

Reviewer #1: Thank you for the opportunity to review the manuscript post revision. I believe the authors have done a great job at addressing my and the other reviewers comments, and implemented necessary changes into the manuscript. i believe this will make a meaningful contribution to the literature, and commend the authors on their focus on fathers who are grossly underrepresented.

Reviewer #2: The authors have responded to all of my concerns and suggestions. I look forward to seeing this in print.

**Do you want your identity to be public for this peer review?** For information about this choice, including consent withdrawal, please see our Privacy Policy

Reviewer #1: **Yes: ** Sarah J. Manuele

Reviewer #2: No

---

## [Editor Report · Acceptance letter]

PONE-D-24-38282R1

PLOS ONE

Dear Dr. Gadian,

I'm pleased to inform you that your manuscript has been deemed suitable for publication in PLOS ONE. Congratulations! Your manuscript is now being handed over to our production team.

Kind regards,

on behalf of

Dr. Ayse Ulgen

Academic Editor

PLOS ONE